

# First evidence of sexual dimorphism in olfactory organs of deep-sea lanternfishes (Myctophidae)

Rene P. Martin[1,2] and W. Leo Smith[1]

[1] Department of Ecology and Evolutionary Biology and Biodiversity Institute, University of Kansas, Lawrence, Kansas, United States
[2] Division of Ichthyology, American Museum of Natural History, New York, New York, United States

## ABSTRACT

Finding a mate is of the utmost importance for organisms, and the traits associated with successfully finding one can be under strong selective pressures. In habitats where biomass and population density is often low, like the enormous open spaces of the deep sea, animals have evolved many adaptations for finding mates. One convergent adaptation seen in many deep-sea fishes is sexual dimorphism in olfactory organs, where, relative to body size, males have evolved greatly enlarged olfactory organs compared to females. Females are known to give off chemical cues such as pheromones, and these chemical stimuli can traverse long distances in the stable, stratified water of the deep sea and be picked up by the olfactory organs of males. This adaptation is believed to help males in multiple lineages of fishes find mates in deep-sea habitats. In this study, we describe the first morphological evidence of sexual dimorphism in the olfactory organs of lanternfishes (Myctophidae) in the genus *Loweina*. Lanternfishes are one of the most abundant vertebrates in the deep sea and are hypothesized to use visual signals from bioluminescence for mate recognition or mate detection. Bioluminescent cues that are readily visible at distances as far as 10 m in the aphotic deep sea are likely important for high population density lanternfish species that have high mate encounter rates. In contrast, myctophids found in lower density environments where species encounter rates are lower, like those in *Loweina*, likely benefit from longer-range chemical or olfactory cues for finding and identifying mates.

## INTRODUCTION

The pelagic deep sea, encompassing open-ocean areas deeper than 200 m, is one of the largest and most stable habitats on Earth. This environment is characterized by a combination of relatively static yet extreme abiotic factors unlike near-shore or terrestrial habitats. This unique combination includes cold temperatures (*Millero, 2005*), stable stratified water layers (*Li et al., 2020*), and little to no sunlight (*Clarke & Backus, 1964*). In this realm, organismal biomass diminishes with depth (*Herring, 2000*), and the fishes found in the sparsely populated deep sea are tasked with adapting to the increased

Corresponding author
Rene P. Martin,
lampichthys@gmail.com

difficulty of locating mates. This contrasts sharply with the higher animal density and smaller habitats (*i.e.*, coral reefs, streams and ponds, brackish waters) of near-shore marine or freshwater environments, where varied niches and higher encounter rates with conspecifics are more common. Deep-sea pelagic fish species may also have increased difficulty finding mates relative to many other habitats because fishes in these environments do not generally school or aggregate at spawning grounds (*Sadovy De Mitcheson et al., 2008*) or possess mating-site fidelity (*Thorstad et al., 2008*). In order to find mates in near-shore and freshwater habitats, where population density is often higher but habitat conditions are more complex than those of the deep sea, fishes have evolved a variety of modifications to their sensory systems to pick up specific sensory cues. In well-lit clear-water habitats, many fish species possess sex-specific color patterns thought to be used as visual cues for the recognition of conspecifics (*Maan et al., 2004*). Olfaction and olfactory cues associated with discrete streams allow salmon to detect their birth/spawning stream (*Stewart et al., 2004*). In species that occur in higher abundances, like the Brown Surgeonfish (*Acanthurus nigrofuscus* (Forsskål, 1775)), social interactions and cues along migratory paths allow for the simultaneous arrival of populations at daily spawning grounds (*Mazeroll & Montgomery, 1998*). The sensory systems and associated cues that help mediate reproductive behavior in these near-shore or freshwater fishes are as diverse as the habitats that they live in. In contrast, the relatively stable and unique environment of the deep sea, characterized by chemically stable water and the lack of sunlight, has led to the evolution of specialized sensory adaptations for living and reproducing in this habitat. A prime example is bioluminescence, or the production of light by a living organism, an adaptation that has become integral to many pelagic deep-sea species.

Bioluminescence has been a focus of many investigations regarding how fishes find and identify mates in the deep sea (*Herring, 2007*; *Davis, Sparks & Smith, 2016*; *de Busserolles et al., 2015*). The dark, calm, and open waters of the deep sea provide ideal conditions for the use of bioluminescence. Numerous deep-sea organisms have evolved the ability to create and visualize bioluminescent light (*Herring, 2007*; *Haddock, Moline & Case, 2010*); this suggests that being able to produce and communicate with light in the dark deep sea is likely important for a variety of behaviors, including reproduction. Bioluminescence is thought to be important in these behaviors because many species possess sexually dimorphic bioluminescent light organs and filters in their eyes, and many are thought to participate in bioluminescent courtship displays (*Mensinger & Case, 1990*; *Mensinger & Case, 1997*; *Herring, 2007*; *de Busserolles et al., 2015*). Vision, while useful, is limited by distance (*Herring, 2000*). Despite bioluminescent signals being visible at approximately 10 m, the low abundance levels of many deep-sea fishes suggests that this level of irradiance may not effectively serve as a visual cue for mate recognition and detection across broad stretches of the open ocean. The olfactory system presents itself as a viable alternative for mate recognition and detection over longer distances (*Dittman & Quinn, 1996*; *Stewart et al., 2004*; *Mitamura et al., 2005*). This is particularly true in aquatic environments where the slow diffusion of chemical cues enables their detection at large distances (*Kasumyan, 2004*).

Fishes exhibit multiple olfactory-mediated behaviors in response to the detection of chemical stimuli (*e.g.*, *Scholz et al., 1976*; *Pavlov & Kasumyan, 1990*; *Kasumyan, 2004*; *Passos et al., 2013*). Selective pressures on a fish's ability to respond appropriately to these olfactory cues have resulted in the evolution of olfactory organs that are capable of detecting a variety of molecules in aquatic environments, including amino acids, amines, and steroidal compounds (*Hara, 1994*; *Kasumyan, 2004*). Fishes respond to chemical cues like pheromones (*Hara, 1994*), alarm signals (*Bairos-Novak, Ferrari & Chivers, 2019*), or prey-emitted stimuli (*Dixson, Pratchett & Munday, 2012*) through detection *via* the olfactory organs (*Sorensen & Baker, 2014*). Across fishes, the main olfactory organs vary in size and shape, are paired, and are located anteriorly on the head, sitting in membranous olfactory chambers with incurrent and excurrent nares (*Kasumyan, 2004*). In many fish species, these organs are made up of olfactory epithelia forming the shape of a rosette composed of lamellae that support different types of olfactory receptor neurons and their supporting cells (*Kasumyan, 2004*). It is generally expected that with a larger olfactory organ and with a larger number of olfactory lamellae within the organ that there will be a concomitant increase in olfactory receptor area if not sensitivity, as more lamellar area provides a greater surface area for chemical stimuli to be detected (*Kasumyan, 2004*), but this is not always the case (*Blin et al., 2018*). It is additionally believed that a greater surface area of the olfactory organ can also lead to being able to discern more types of molecules and may also be correlated with having additional types of olfactory sensory neurons (*Policarpo et al., 2021*, *2022*). These olfactory organs are an important system used in mediating a variety of behaviors in fishes. Empirical studies have shown fishes use olfactory cues in prey detection and feeding (*Johannesen, Dunn & Morrell, 2012*) or for sensing predators (*Dixson, Pratchett & Munday, 2012*). Fishes also use olfaction to discriminate between water masses, to orient themselves for settlement, and to detect spawning grounds (*Atema, Kingsford & Gerlach, 2002*; *Stewart et al., 2004*). Additionally, fishes use olfactory cues to identify mates (*Boyle & Tricas, 2014*) and to assess spawning readiness in conspecifics (*Hara, 1994*). Chemical signals can travel large distances (particularly relative to sight and mechanoreception), and the distance these signals can be detected is determined by chemical concentration, speed of diffusion, and molecular decay (*Jumper & Baird, 1991*). In the deep sea, these chemicals can disperse within a stratified layer of water without much diffusion.

Sexual dimorphism of olfactory-organ morphology has rarely been observed in fishes outside of deep-sea benthic (Halosauridae, Lophiidae) and deep-sea pelagic (Ceratiidae, Cetomimidae, Eurypharyngidae, Gonostomatidae, Sternoptychidae) fishes (*Marshall, 1967*; *Marshall, 1971*; *Marshall, 1979*; *Caruso, 1975*; *Gibbs, 1991*; *Kasumyan, 2004*; *Pietsch, 2005*; *Johnson et al., 2009*). *Marshall (1967)* hypothesized that in species where sexual dimorphism occurs, the olfactory organs are being used for mate recognition or detection. In many of the deep-sea fish species that possess sexual dimorphism in olfactory-organ morphology, the male has significantly enlarged olfactory organs compared to the female. In many of these species, there is also dimorphism in body size, with males being significantly smaller than females. This phenomenon is evident in certain deep-sea

anglerfish species, such as *Cryptopsaras couesii* Gill, 1883, where males, measuring merely 40 mm in standard length (SL), are found attached to females that are significantly larger, reaching lengths beyond 400 mm SL (*Pietsch, 2005*). These males, prior to attachment, may have olfactory organs with a length approximately 10% of their SL, whereas olfactory organs in females may only be about 1% of their SL. Until recently, the extreme sexual dimorphism documented between male and female whalefishes (Cetomimidae) resulted in them being placed into different, closely related families (*Johnson et al., 2009*). Males, previously placed in the Megalomycteridae, or the 'bignose fishes,' are smaller than females and possess significantly enlarged olfactory organs relative to their body size compared to females. Although there have been no studies investigating male cetomimids using these enlarged olfactory organs to find females, models of mate encounter rates using chemical signaling have been generated for the mesopelagic (open ocean between 200–1,000 m) sternoptychid *Argyropelecus hemigymnus* Cocco, 1829. In this species, males possess greatly enlarged olfactory organs compared to females, and *Baird & Jumper (1995)* suggested that pheromone release by females and detection by males greatly increases the potential for finding mates in the immense mesopelagic zone. Chemosensory systems are thought to be especially important for sensing predators and finding prey for fishes living in these large deep-sea expanses (*Gibbs, 1991*), yet in many of these sexually dimorphic species, males either cease to feed once becoming sexually mature, as in whalefishes (*Johnson et al., 2009*), or males and females possess similar diets as seen in the sternoptychid *A. hemigymnus* (*Carmo et al., 2015*), indicating that variation in feeding is unlikely to result in the observed sexual dimorphism in their olfactory organs.

Studies on the olfaction of fishes that live in near-shore and freshwater habitats, where chemical cues can be empirically tested, are important for understanding fish behavior tied to olfaction (*Caprio, 1978*, *1980*; *Cardwell, Dulka & Stacey, 1992*; *Hara, 2006*). Discerning the relationship between these factors and olfactory-organ morphology is especially important for analyzing deep-sea fishes where, due to issues related to fish behavior, habitat type, depth, and sampling techniques, it can be difficult to observe them *in situ* or keep specimens alive in a lab. Thus, the examination of olfactory organs in deep-sea fish species has remained relatively rare and our current knowledge of the frequency and breadth of sexual dimorphism in the olfactory organs across deep-sea fishes is likely to be an underestimate. In light of a broader study analyzing olfactory-organ anatomy and morphology across 105 of the 252 lanternfish species, we discovered and herein describe the first case of sexual dimorphism of the olfactory organ in two closely related species of lanternfishes (Myctophidae).

## MATERIALS AND METHODS

### Specimens

Following a broader study on lanternfish olfactory-organ morphological variation that examined 105 species across all genera of lanternfishes (in prep.), we identified sexual dimorphism in one genus of myctophids. Sexual dimorphism was not seen in the olfactory organs in other genera of lanternfishes, but in order to better understand the extent of this trait within a phylogenetic context, we performed a more detailed examination of the gross

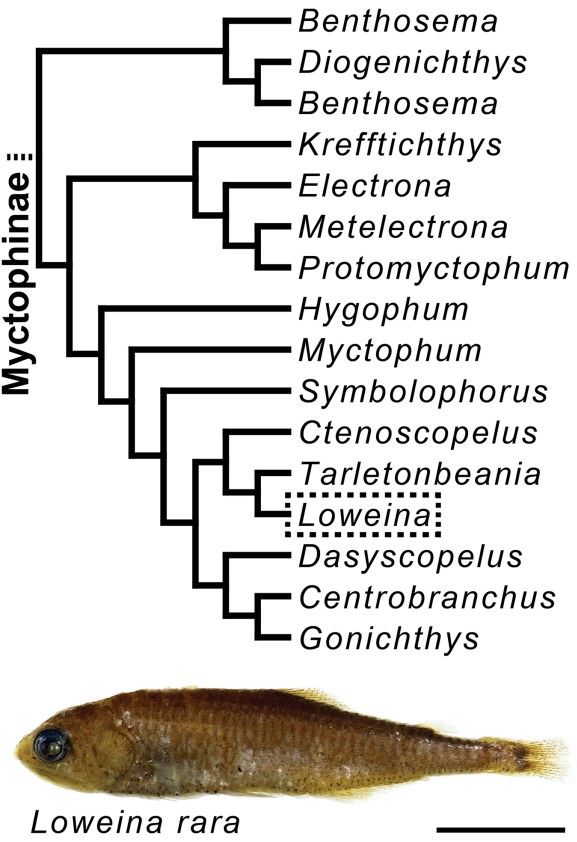

*Loweina rara*

**Figure 1 Myctophine phylogenetic relationships and preserved specimen of *Loweina rara*.** Trimmed lanternfish phylogeny showing some generic-level relationships in the subfamily Myctophinae from *Martin et al. (2018)* using a combination of ultra-conserved elements, Sanger sequence data, and morphological characters. Specimen of *Loweina rara* (MCZ 151184). Scale bar represents 1 cm.

morphology of the olfactory organs in 72 specimens from six lanternfish species, with taxonomic sampling efforts focused on species in *Loweina* (*L. interrupta* (Tåning, 1928), *L. rara* (Lütken, 1892), and *L. terminata* Becker, 1964) and allied genera, including *Ctenoscopelus phengodes* (Lütken, 1892), *Tarletonbeania crenularis* (Jordan & Gilbert, 1880), and *T. taylori* Mead, 1953 (Fig. 1), a clade hypothesized to have evolved approximately 17 million years ago, and with *Loweina* and *Tarletonbeania* diverging around nine million years ago (*Denton, 2018*). All specimens were formalin or ethanol fixed and alcohol (ethanol or isopropanol) preserved museum specimens from the Museum of Comparative Zoology, the Natural History Museum of Los Angeles County, the National Museum of Natural History, or the Scripps Institution of Oceanography. A full list of specimens used in this study can be found in Table S1. Museum codes follow *Sabaj (2020)*.

## Gross morphology

Adult male and female specimens from each species were examined except for *Loweina terminata*, where only male specimens were available. Specimen numbers per

species ranged from 5–23. The standard length of specimens was measured and sex was determined non-invasively using the presence/absence of unique sex-specific morphologies of sexually dimorphic light organs. Olfactory-organ rosettes were measured and the lamellae of both the left and right organ were counted using a Unitron Z8 Series stereomicroscope. Olfactory-organ rosettes were quantified using two length measurements: the major axis (the distance from end-to-end measured along the rosette midline) and the minor axis (the distance from side-to-side measured along a line normal to and bisecting the major axis) as described by *Baird & Jumper (1993)*. Measurements were made to the nearest 0.01 mm using a Mitutoyo dial caliper. Given that olfactory lamellae collapse slightly outside of liquid and that they have a club-like shape, it is difficult to determine their exact size, so length and width measurements were made to obtain a relative indication/approximation of differences in the inter- and intraspecific variation in organ size.

### Statistical analyses

In order to analyze the differences between left-side and right-side olfactory-organ lengths and widths and also between body sizes of males and females of species in *Loweina*, t-tests were run using the 't.test' function from the base *stats* package in R (*R Core Team, 2022*). Linear regressions by species were performed using the 'lm' function also from the base *stats* package. Analyzing relationships between morphological traits (olfactory organ length, lamellar counts, and standard length) and sex, ANCOVAs were performed for each species using the function 'anova_test' from the *rstatix* package (*Kassambara, 2021*). These ANCOVAs were used to test whether olfactory-organ length and lamellar counts differ between sexes in lanternfish species while correcting for the effects of standard length. Plots were created using the 'ggscatter' function from the package *ggpubr* and *ggplot2* (*Kassambara, 2020*). The R code used for analyses and associated raw data files can be found in the Supplemental Files.

## RESULTS

All examined lanternfishes possess closely set incurrent and excurrent nares (Fig. 2A) and olfactory rosettes (Fig. 2B) that are round-to-oval shaped and oriented rostrocaudally in the olfactory chamber. Lamellae are club/paddle shaped, with broad/compressed distal ends and narrower proximal ends attached to the central raphe. Lamellae are attached to the raphe at approximately a 90° angle except for the posteriormost lamellae, which are evenly spaced and fan radially around the end of the raphe (Fig. 2B). Lamellae are mostly oriented perpendicular to water flow. Generally, the entire olfactory rosette is attached to the basal epithelium of the olfactory chamber *via* the raphe, with the smallest lamellae at the most anterior end of the rosette and the largest lamellae at the most posterior end (Fig. 2B). Left- and right-side olfactory organs are not significantly different in size ($p = 0.7885$) or number of lamellae ($p = 1$), thus left-side organ measurements were used in downstream analyses. There is variation in lamellar counts and organ size among individuals of the same species and among different species (Fig. 3; Table 1). There is a small but significant increase in the approximate size of the olfactory organ with increasing

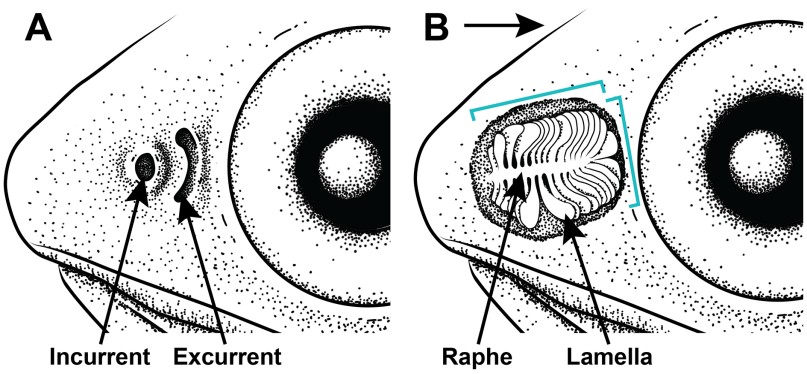

**Figure 2 Illustration of the gross morphology of lanternfish nares and olfactory organ.** Generalized illustration of the gross morphology of lanternfish nares and olfactory organs. (A) Incurrent and excurrent nares. (B) Nares and surrounding tissue dissected away exposing olfactory organ. Arrow pointing in the direction of water flow and brackets depicting approximate length and width measurements.

**Figure 3 Statistical analyses for olfactory-organ size by specimen length in six species of lanternfishes.** Linear regression analyses for lanternfish specimen standard length on olfactory-organ length and number of lamellae. Boxplots showing standard length-corrected lamellar counts between male and female individuals of species of *Loweina*.

**Table 1 Results from the statistical analyses on olfactory organ length and lamellar counts in multiple species of lanternfishes.**

| t-test | p | dF | t | |
|---|---|---|---|---|
| Left/Right organ length | 0.7885 | 71 | 0.2693 | |
| Left/Right lamellar counts | 1 | 71 | 0 | |
| SL between males and females *Loweina interrupta/Loweina rara* | 0.2861/0.757 | 16.9/4.2 | 1.10/−0.33 | |
| **Linear regression-organ length/lamellar counts** | **p** | **dF** | **F** | **Adjusted R²** |
| Species | | | | |
| *Ctenoscopelus phengodes* | **7.523e−05/0.001633** | 8 | 54.96/21.68 | 0.857/0.6967 |
| *Loweina interrupta* | 0.2693/0.9351 | 7 | 1.439/0.007124 | 0.0521/−0.1417 |
| *Loweina rara* | 0.1065/0.9083 | 21 | 2.844/0.01361 | 0.07734/−0.04694 |
| *Loweina terminata* | 0.3390/0.4397 | 3 | 1.287/0.7895 | 0.06699/−0.05556 |
| *Tarletonbeania crenularis* | **0.0009787/0.1948** | 13 | 17.91/1.868 | 0.5471/0.05841 |
| *Tarletonbeania taylori* | **0.005507/0.00681** | 8 | 14.17/13.08 | 0.5941/0.5731 |
| **ANCOVA (Sex/SL)-organ length/lamellar counts** | **p** | **dF** | **F** | |
| Species | | | | |
| *Ctenoscopelus phengodes* | 0.391/0.403 | 6 | 0.692/1.06 | |
| *Loweina interrupta* | **0.031/0.014** | 6 | 7.863/11.602 | |
| *Loweina rara* | **4.79e−06/6.12e−11** | 20 | 38.304/157.538 | |
| *Tarletonbeania crenularis* | 0.333/0.953 | 12 | 1.02/0.004 | |
| *Tarletonbeania taylori* | 0.484/0.997 | 7 | 0.546/0.0000112 | |

Note:
Results from the statistical analyses including linear regressions and ANCOVAs on olfactory organ length and lamellar counts associated with both sex and lanternfish standard length. Bolded $p$ values are significant.

standard length in *Ctenoscopelus* ($p = 7.523e{-}05$) and both species in *Tarletonbeania* ($p = 0.0009787$ for *T. crenularis* and $p = 0.005507$ for *T. taylori*; Fig. 3). This trend is not seen in species of *Loweina* ($p$ values greater than 0.05). There is a small but significant increase in the number of lamellae with increasing standard length in *Ctenoscopelus* ($p = 0.001633$) and *Tarletonbeania taylori* ($p = 0.00681$), but this trend is not significant in *T. crenularis* or species of *Loweina* ($p$ values greater than 0.05). Testing for sexual dimorphism in SL between males and females in *Loweina*, we find no significant difference between lengths in either *L. interrupta* ($p = 0.2861$) or *L. rara* ($p = 0.757$; Fig. 3).

After accounting for standard length, evidence for sexual dimorphism in the olfactory organs occurs only in species of *Loweina* (Fig. 3; Table 1). Within *Loweina*, males of *L. interrupta* possess significantly larger olfactory organs ($p = 0.031$) and significantly more lamellae ($p = 0.014$) than females (Figs. 3, 4). The larger olfactory organ and number of lamellae trend in males was also seen in *L. rara* ($p = 4.79e{-}06$ and $p = 6.12e{-}11$, respectively). Species in *Ctenoscopelus* and *Tarletonbeania* did not exhibit differences in olfactory organ length or lamellar counts between sexes (Table 1).

# DISCUSSION

## Sexual dimorphism in olfactory organs of deep-sea fishes

One notable feature of the olfactory organs in several deep-sea fish lineages is the presence of sexual dimorphism (*e.g.*, *Marshall, 1967*; *Caruso, 1975*; *Marshall, 1979*; *Gibbs, 1991*;

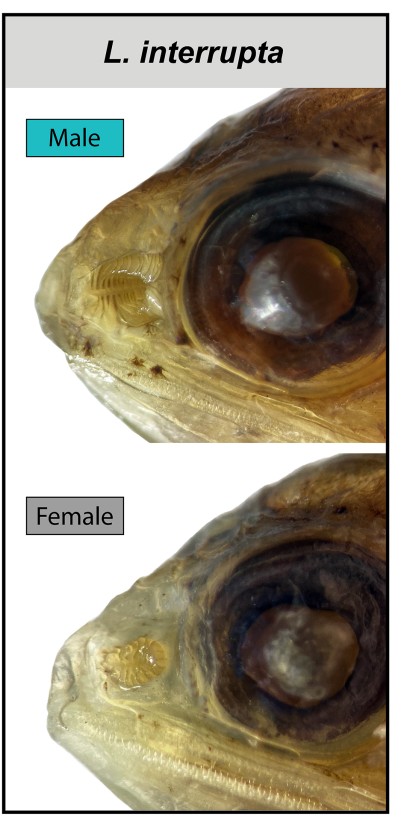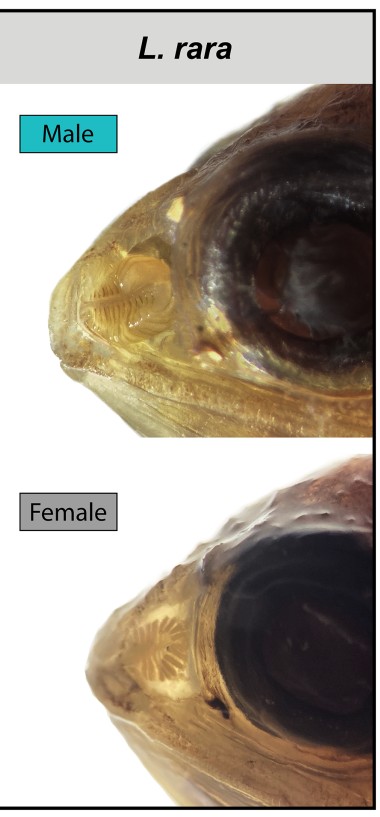

**Figure 4** **Male and female olfactory-organ images of two species of *Loweina*.** Images showing sexual dimorphism in olfactory organs in two species of *Loweina*, *L. interrupta* and *L. rara*. Male *L. interrupta*, 30 mm SL, LACM 11511-19; female *L. interrupta*, 25 mm SL, LACM 34953-1; male *L. rara*, 28 mm SL, SIO 15-959; female *L. rara*, 37 mm SL, MCZ 164348.

*Pietsch, 2005*), an uncommon occurrence in fishes living in other habitats (*Kasumyan, 2004*). In this study, we present evidence for sexual dimorphism in the olfactory organs in two species in the deep-sea lanternfish genus *Loweina*, the first case in the Myctophidae. Male specimens of *L. interrupta* and *L. rara* possess significantly larger olfactory organs than females ($p = 0.031$ and $p = 4.79e{-}06$, respectively) and a significantly higher number of lamellae ($p = 0.014$ and $p = 6.12e{-}11$). Although we were unable to obtain mature female specimens of *L. terminata*, we believe this trend continues in this species, as relative to body size, males have even larger olfactory-organ lengths and higher lamellar counts than *L. interrupta* and *L. rara* (Fig. 3). This finding adds to the growing list of deep-sea fish lineages that possess sexually dimorphic olfactory organs, including groups like the ceratiids (sea devils), eurypharyngids (pelican eels), sternoptychids (deep-sea hatchetfishes), and cetomimids (whalefishes; *Caruso, 1975*; *Marshall, 1979*; *Pietsch, 2005*; *Johnson et al., 2009*). Hypotheses regarding the adaptive significance behind sexual dimorphism in olfactory-organ morphology in these deep-sea groups is often tied to their use in mate detection and location in the deep sea.

Within the ceratioid anglerfishes, sexual dimorphism is expressed in the difference between enlarged olfactory organs in males and smaller olfactory organs in females
(*Bertelsen, 1951*; *Caruso, 1975*; *Pietsch, 2005*). Males and females also possess extreme sexual dimorphism in body size, with females of some species being greater than ten times larger than males (*Pietsch, 2005*). Males of many ceratioid species are 'parasitic' or 'chimeric,' attaching themselves to a female once found and often fusing themselves to her body (*Pietsch, 2005*). Prior to attachment, males are hypothesized to use their well-developed olfactory organs to pick up pheromone cues given off by slow-moving females (*Bertelsen, 1951*; *Munk, 2000*; *Pietsch, 2005*). Once a male finds and fuses to a female, his olfactory organs degenerate rapidly. In the deep-sea pelican eel *Eurypharynx pelecanoides* Vaillant, 1882, males possess an enlarged knob occupying a large portion of the face which contains the olfactory rosette composed of numerous lamellae. Alternatively, females of *E. pelecanoides* of the same size class possess almost imperceptible olfactory organs (*Gartner, 1983*). Males of *E. pelecanoides* exhibiting enlarged olfactory rosettes also possess well-developed testes that fill the majority of their body cavity. *Gartner (1983)* hypothesized that the presence of these extremely enlarged olfactory organs indicated that sexually mature male *E. pelecanoides* are using olfaction for mate detection *via* stimulation from pheromones given off by females. Alongside deep-sea anglerfishes and pelican eels, many bristlemouths (Gonostomatidae) have sexually dimorphic olfactory organs (*e.g.*, *Cyclothone*, *Sigmops*). Compared to females, mature males of *Cyclothone* possess significantly enlarged olfactory organs, nerves, olfactory bulbs, and forebrains (*Marshall, 1967*, *1971*; *Gibbs, 1991*). Specifically, mature males of *C. braueri* Jespersen and Tåning, 1926 (*Badcock & Merrett, 1976*) and some males of *C. pallida* Brauer, 1902 possess olfactory rosettes, whereas there was no evidence of these organs in females (*Maynard, 1982*). Males of species of *Cyclothone* are also thought to possess more well-developed muscles than females of a similar size class, which *Maynard (1982)* hypothesized would facilitate more locomotive males to find more stationary females (*Marshall, 1971*). Additionally, sexual dimorphism has been identified in the olfactory organs of multiple species of sternoptychids (*Argyropelecus hemigymnus*, *Valenciennellus tripunctulatus* (Esmark, 1871); *Baird, Jumper & Gallaher, 1990*). *Jumper & Baird (1991)* hypothesized and modeled pheromone signaling by female sternoptychids and pheromone diffusion theory in stratified deep-sea waters. Their model showed how effective mate detection by olfaction could be in *A. hemigymnus* (*Jumper & Baird, 1991*; *Baird & Jumper, 1995*), a small, abundant, bioluminescent mesopelagic fish that mainly eats zooplankton and possesses traits similar to those of lanternfishes. *Jumper & Baird (1991)* suggested that pheromones released by females diffuse horizontally along stable stratified layers of water to create patches that have a high success rate for long-range detection by searching males. Recent work assessing surface area and number of lamellar folds in the olfactory organ found that, in fishes, greater surface of the olfactory organ was correlated with both an increased ability to discern a larger number of molecules and with having additional types of olfactory receptors (*Policarpo et al., 2021*, *2022*). Repeated instances of sexual dimorphism in the olfactory organs across multiple deep-sea fish lineages lends support to the hypothesis that olfaction is very likely being used for finding mates, especially if searching males with larger olfactory organs have evolved the ability to discern species-specific pheromones given off by females.

 

### *Loweina*

In this study, we find that two of the three species in *Loweina* exhibit significant differences in the number of lamellae and size of olfactory organs between males and females (Figs. 3, 4). Olfactory-organ shape is similar between males and females, but their organs differ significantly in size and lamellar counts (Fig. 4; Table 1). We found no significant trend in increasing organ size with increasing lamellar counts in *Loweina interrupta* or *L. rara* (Fig. 3) due to the inclusion of both males and females in the analysis. In the ANCOVAs, after accounting for the confounding variable of sex, we found statistically significant differences in organ length and lamellar counts between males and females of *L. interrupta* and *L. rara* (Fig. 3; Table 1). Based on our findings, this adaptation likely occurs in *L. terminata* as well, but we were unable to obtain mature female specimens for analysis (Fig. 3).

Species in *Loweina* are small, with adults not usually reaching lengths beyond 45 mm SL. Additionally, they possess sexually dimorphic caudal light organs, similar to many other lanternfish species in the subfamily Myctophinae (*Wisner, 1976*; *Martin, Davis & Smith, 2022*). Males exhibit greatly enlarged light organs on their tails, which are absent in females. Other than these caudal light organs, males and females in species of *Loweina* are superficially similar in their morphology (*Nafpaktitis et al., 1977*; *Martin, Davis & Smith, 2022*). They are one of the few groups of lanternfishes that possess subterminal mouths, along with species in *Centrobranchus* and *Gonichthys* (*Nafpaktitis et al., 1977*). *Loweina rara* reaches maturity between 28–32 mm (*Nafpaktitis et al., 1977*) and, similar to other lanternfishes, possesses planktonic eggs and larvae (*Moser & Ahlstrom, 1996*). Their larval morphology is unusual among lanternfishes in that they possess an elongate and ornamented lowermost pectoral-fin ray (*Moser & Ahlstrom, 1970*; *Evseenko et al., 1998*). Elongated filaments and appendages on larval fishes are hypothesized to be used in predator deception (*Govoni et al., 1984*; *Greer et al., 2016*). Unfortunately, there is limited knowledge on the larval morphology in species of *Loweina* which is caused, in part, due to their scarcity in plankton samples (*Evseenko et al., 1998*).

Relative to many other lanternfish species, less is known about the ecology of species in *Loweina*. *Loweina interrupta* occurs in northern and southern temperate waters worldwide, *L. rara* is known to occur in the mesopelagic zone between 25° and 45° latitude in both hemispheres but is uncommon, and *L. terminata* is only found in the temperate North Pacific Ocean (*Evseenko et al., 1998*). Like many other lanternfishes, species in *Loweina* are vertical migrators, and, around Bermuda, *L. rara* is distributed from 150–300 m at night and 800–1,000 m during the day (*Gibbs & Krueger, 1987*).

### Comments on mate detection in *Loweina*

Lanternfishes are bioluminescent, and their species-specific and sexually dimorphic light organs are thought to be used for intraspecific communication and species recognition (*Herring, 2007*; *Martin, Davis & Smith, 2022*). As seen in many other lanternfishes, species of *Loweina* possess sexually dimorphic caudal light organs, where males possess a large supracaudal gland while females lack caudal light organs (*Paxton, 1972*; *Martin, Davis & Smith, 2022*). Sexually dimorphic bioluminescent light organs and signals have evolved

numerous times in deep-sea fishes (*Herring, 2007*). The dark open vistas of the deep sea that lack the complexity of visual cues found in near-shore and freshwater habitats provide an idealized setting for communication *via* light signals. Vision and visual cues like bioluminescence are hypothesized to be useful for identifying mates and conspecifics among fishes living in the deep sea (*Paxton, 1972*; *Davis et al., 2014*), but depending on the intensity of the light, a visual signal in the ocean may only travel approximately ten meters (*Herring, 2000*). This limited distance is not an issue for highly abundant species that have high encounter rates but can be a problem for species that have low population density. Lanternfishes are one of the most abundant vertebrate groups on the planet (*Olivar et al., 2012*). Many species are known to aggregate and have little trouble meeting the bioluminescent visual criteria of being within at least ten meters (*Flynn & Paxton, 2012*). In contrast, species of *Loweina* are much scarcer and are less abundant than other lanternfishes (*Wisner, 1976*; *Nafpaktitis et al., 1977*). Bioluminescent signaling as a means of mate detection *via* a visual sensory system in the more disparately spaced species of *Loweina* might not be feasible, and bioluminescence may instead be used in conjunction with additional sensory systems, in this case, olfaction.

Olfactory-mediated signaling to males *via* pheromones given off by females may be the initial step in mate detection in *Loweina*. Studies on sex pheromones in fishes are abundant, and research suggests that fishes may be able to give off and discriminate species-specific pheromone complexes (*Levesque et al., 2011*; *Sorensen & Baker, 2014*) or even possess species-specific pheromones (*Lim & Sorensen, 2011*; *Kamio, Yambe & Fusetani, 2022*). If a searching male in *Loweina* picks up species-specific pheromones (or pheromone complexes) given off by a female, he may swim toward her *via* an increasing chemical concentration gradient (*Jumper & Baird, 1991*). Once near enough, a male could flash and signal a female using his caudal light organ at a more intermediate and visible range. One problem searching males need to overcome is the initial detection of a female pheromone patch. Both *Loweina interrupta* and *L. rara* are known to migrate either fully or partially in the water column (*Nafpaktitis et al., 1977*; *Gibbs & Krueger, 1987*). If males of these species are using olfactory cues to detect females, vertical movement in the water column could enhance the encounter rate of pheromone patches if they are being given off by females, increasing the opportunity for mate location. Currently, it remains unknown whether distinct differences exist in the migration and movement patterns between males and females. Sexual dimorphism in the olfactory organs of species in *Loweina* may be an indicator that olfaction is being used in species recognition and mate detection in this clade of rare lanternfishes. Additional work finding and documenting the population dynamics of the rare species of *Loweina*, documenting and analyzing olfactory-mediated reproductive behavior, and describing their spawning strategies will be imperative for explaining why sexual dimorphism exists in the olfactory-organ morphology of species in this genus. Findings from future work may lend evidence of the use of olfaction in the reproductive strategy of these relatively rare species of *Loweina*.

## Diet, prey selectivity, and olfaction

In addition to playing a role in mediating reproductive behaviors, olfactory organs are part of the well-documented chemosensory system fishes use to pick up chemical cues associated with feeding (*e.g.*, *McBride et al., 1962*; *Hara, 1994*). Chemosensory systems are thought to be especially important for sensing predators and finding prey for fishes living in large deep-sea expanses (*Gibbs, 1991*). Although olfaction is important for picking up chemical cues associated with feeding in many fishes, we believe the sexually dimorphic differences in olfactory-organ morphology in *Loweina* are not associated with feeding behaviors. Broadly, lanternfishes are zooplanktivores and most species feed on abundant zooplankton in the epipelagic (*e.g.*, *Pakhomov, Perissinotto & McQuaid, 1996*; *Oliva, Ulloa & Bleck, 2006*; *Sassa & Takasuka, 2020*). Lanternfishes are believed to be vision-based feeders (*de Busserolles et al., 2013*). Studies on their diets suggest that they are generally opportunistic feeders and that their diets are also often associated with prey availability in their specific geographic location (*Sameoto, 1988*; *Kozlov, 1995*). Prey selection in lanternfishes has also been tied to mouth gape or tooth adaptations, as many species are known to shift their diet as they grow in size to incorporate a broader range of prey sizes, and in some species, special heterodont dentition is thought to help keep prey in the mouth (*e.g.*, *Hopkins, Sutton & Lancraft, 1996*; *Williams et al., 2001*; *Conley & Hopkins, 2004*; *Shreeve et al., 2009*; *Tanaka et al., 2013*; *Martin & Davis, 2020*). There is only one known case of specialized feeding in lanternfishes, which occurs in *Centrobranchus* (*Hopkins & Gartner, 1992*). All three species of *Centrobranchus* specialize on pelagic hard-bodied cavoliniid pteropods, and these lanternfishes also possess modified crushing plates on their gill rakers to consume this specialized prey item (*Hopkins & Gartner, 1992*). Shifts in prey type and feeding based on variables such as geography and size suggest that other than for basic chemical cues, the majority of lanternfishes are not relying on specialized chemical cues and olfaction for prey location. Unfortunately, most diet studies do not assess any sex-based prey selectivity. Despite the abundance of publications on the diets of different lanternfish species, we failed to find any that include species of *Loweina*, likely due in part to their rarity compared to other lanternfish species.

Additional evidence suggests that, among the diversity of deep-sea fish lineages with sexual differences in their olfactory organs, males use their larger olfactory organs for activities beyond feeding. In at least one species of sternoptychid (*Argyropelecus hemigymnus*) with sexually dimorphic olfactory organs, all specimens studied had similar diets (*Carmo et al., 2015*), indicating that variation in feeding is unlikely to result in observed sexually dimorphic differences in their olfactory morphologies. In whalefishes, males possess greatly enlarged olfactory organs relative to body size compared to females, but the evidence suggests that adult males cease feeding after becoming sexually mature and lose both their stomachs and esophagi (*Johnson et al., 2009*). Additionally, *Bertelsen (1951)* and *Pietsch (2005)* suggested that many deep-sea anglerfish jaw and tooth morphologies in mature males are ill-equipped for prey capture. Further, they noted that many male anglerfishes lack a fully developed alimentary canal, making it unlikely that

mature males would be using olfaction to find prey. Given that the mature males in many lineages of deep-sea fishes that have sexually dimorphic olfactory organs do not feed and the lack of specialized diets among most lanternfish species (see above), we do not believe the sexual dimorphism of the olfactory organs in *Loweina* is associated with prey discovery. Future work focused on gut content analysis between males and females of species in *Loweina* would provide insight not only into the feeding preference in this understudied and rare genus of lanternfishes, but finding similarities in diets between sexes would also lend support to the hypothesis that olfaction is not being used differentially between males and females to find prey.

## CONCLUSIONS

Herein we describe the first instance of sexual dimorphism in the olfactory organs of lanternfishes. This lanternfish study adds to the growing understanding of deep-sea fish lineages that possess sexually dimorphic olfactory organs (*Badcock & Merrett, 1976*; *Gartner, 1983*; *Pietsch, 2005*; *Johnson et al., 2009*). We report that males of two lanternfish species in the genus *Loweina*, *L. interrupta* and *L. rara*, exhibit significantly enlarged olfactory organs compared to females (Figs. 3, 4). The male morphology with increased lamellar counts suggests that this dimorphism is also found in the third species of *Loweina*, *L. terminata*, but we were unable to obtain sexually mature females for analysis in this study. Further, we show that the close relatives of *Loweina* do not possess sexually dimorphic olfactory-organ morphologies such that this feature is restricted to this genus. Historically, lanternfishes have been hypothesized to emphasize visual orientation in their predation and communication (*Davis et al., 2014*; *de Busserolles, Marshall & Collin, 2014*; *de Busserolles et al., 2015*). The presence of sexually dimorphic light organs exhibited by numerous lanternfish species, including species of *Loweina*, indicates the importance of this adaptation in mate recognition or mate detection in this group. The interplay of vision and bioluminescence undoubtedly plays an important role in communication and mate recognition at intermediate distances of up to approximately 10 m (*Herring, 2000*), and many lanternfish species are extremely abundant and form large aggregations (*Flynn & Paxton, 2012*), increasing the likelihood of encountering conspecifics. In relatively scarce species, like those of *Loweina*, where the chance of encountering a mate in the deep sea may be low, the use of bioluminescent signals may come after the use of chemical cues like pheromones, and these olfactory signals can be detected from farther distances. A male individual of *Loweina* possessing enlarged and potentially more sensitive olfactory organs may seek and detect chemical cues given off by a female and navigate toward her. Once the male is within her visual distance, he may use his sexually dimorphic light organ to signal her with a bioluminescent display. Sensory processing for sexual encounters in the deep sea can be a challenge, especially for species with low population abundances. Finding a mate in these vast open-ocean areas may require the integration of multiple sensory systems, as the evidence suggests *Loweina* are using at least olfaction and vision.

## ACKNOWLEDGEMENTS

We would like to thank following people and institutions for providing specimens used in this study: T. Clardy, B. Ludt (LACM); A. Williston, M. Sorce (MCZ); D. Arcila, B. Frable (SIO); L. Parenti, D. Pitassy (USNM). We would additionally like to thank Dr. Waiho, Dr. Policarpo, Dr. Sutton, and one anonymous reviewer for their helpful and insightful comments on our research.

### Funding

Funding for this work was provided by the University of Kansas Ecology and Evolutionary Biology Summer Research Award, the University of Kansas Biodiversity Institute Panorama Grant, and the American Museum of Natural History Lerner-Gray Marine Research Grant. The funders had no role in study design, data collection and analysis, decision to publish, or preparation of the manuscript.

### Grant Disclosures

The following grant information was disclosed by the authors:
University of Kansas Ecology and Evolutionary Biology Summer Research Award.
University of Kansas Biodiversity Institute Panorama Grant.
American Museum of Natural History Lerner-Gray Marine Research Grant.

### Competing Interests

The authors declare that they have no competing interests.

### Author Contributions

- Rene P. Martin conceived and designed the experiments, performed the experiments, analyzed the data, prepared figures and/or tables, authored or reviewed drafts of the article, and approved the final draft.
- W. Leo Smith analyzed the data, authored or reviewed drafts of the article, and approved the final draft.

### Data Availability

The 74 museum specimens used, their associated museum ID's, and their raw measurements and the annotated R code used for statistical analyses on olfactory organs are available in the Supplemental Files.

### Supplemental Information

Supplemental information for this article can be found online at http://dx.doi.org/10.7717/peerj.17075#supplemental-information.

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
