# Peer review of "First evidence of sexual dimorphism in olfactory organs of deep-sea lanternfishes (Myctophidae)"

_PeerJ, doi:10.7717/peerj.17075_

## Round 0.1 · original submission · Major Revisions

I agree with the reviewers that the overall framework of this research is thorough and detailed, and explains a very important topic (sexual dimorphism in olfactory organs) of lanternfish. The revisions suggested by the three reviewers are not extensive, and I look forward to reading the revised version of the manuscript soon.

Reviewer 1 ·

Basic reporting

The manuscript by Martin & Smith investigates the sexual dimorphism in the olfactory organs of lanternfishes, and relates them with their unique mating strategy in the deep sea. The Introduction beautifully describes the needed background of the research, providing sufficient information for readers before moving on to the later sections. The research framework designed by the authors sufficiently answers their postulate, with samples not only from a single species, but drawing conclusion from the analyses of six species across three genera. The method is quite straightforward – measure the size of the olfactory-organ and compare between sexes.

Experimental design

I would also expect the authors to check for sexual dimorphism in terms of body size, and see if sexual dimorphism of the olfactory organ is also showing the same or opposite pattern with the sexual dimorphism patterns of body size.

Also, would it be possible to conduct further microscopy description of the formalin or ethanol fixed samples? Such as SEM? Or other advanced microscopy techniques?

Validity of the findings

Some parts of the Results, such as line 191-199 will be easier to understand if the authors provide some figures to these descriptions.

Figure 3 illustrates the olfactory organ of Loweina. But I think a real picture of this olfactory organ will be extremely helpful, especially to highlight the sexually dimorphic characters between males and females of different species. The authors should have these photographs as they measure the length of the olfactory-organ rosettes under a stereomicroscope. Providing these pictures will also help readers to understand the morphology of this organ.

·

Basic reporting

In this manuscript, Rene Martin and William Smith examine the olfactory epithelium morphology of six lantern fish species. They found substantial differences in the epithelium size and in the number of lamellae between species investigated, ranging from 10 to more than 40. Overall, there was only slight correlations between the length of a species and the number of lamellae, which is unexpected and very interesting. Furthermore, they observed that males had significantly more lamellae than females when taking the standard length into account. The authors make the hypothesis that this is not due to diet but most likely due to the reproductive mode of these species, where the male would need to find the female who releases pheromones.

Overall, I find the manuscript well written and very interesting. I have few comments which would need to be addressed. The most important thing to me is to add pictures of the olfactory epithelium investigated in this study. It also appears important to me to make a clear distinction between olfactory sensitivity and discriminating power, which can be both increased with the number of lamellae. This could be mentioned in the introduction but also in the discussion.

Materials & Methods

Line 29: specify that m = meters

Line 98: I would also say that an increase of the number of lamellae can also lead to a larger repertoire of odorant molecules that can be recognized. This is support by recent genetic studies where they show that species with more lamellae have more different olfactory receptors:

1) https://doi.org/10.1186/s12915-022-01397-x
2) https://doi.org/10.1093/molbev/msab145

I think it would also be important to mention more recent studies on the relation between the olfactory epithelium size and the olfactory sensitivity. There is an example in Astyanax mexicanus where authors show that there is not a strict correlation between the olfactory epithelium size and the olfactory sensitivity.

1) https://doi.org/10.1016/j.ydbio.2018.04.019



Line 169: Based on my experience, most of the time, the olfactory epithelium is not flat, and lamellae are folded. This, in combination with its fragility, I had a really hard time to measure its length. Did you have the same issue, and if yes, can you precise if the length you report are approximations or real lengths?


One thing very important to me and which is missing in the manuscript and associated file are pictures of the olfactory epithelium you used in this study. At-least a representative set of pictures from different individuals and different species with different lamellae number. If you cannot provide those pictures, please tell why.


Results

The results and discussion are well written and are clear and correct to me.


Figures

Figure 1: Do you have any estimations of the divergence time between these different species? If yes it could be nice to have them in the figure.

Figure 2: Could you please add a third plot, with the relation between the olfactory epithelium length and the number or lamellae?

Figure 3: Replace that by real pictures of olfactory epithelium please.

Experimental design

no comment

Validity of the findings

As said previously (1- Basic reporting), the authors need to provide olfactory epithelium pictures.

·

Basic reporting

This article is very well-written, with a thorough literature review. I was actually surprised that sexual dimorphism in myctophids had not been reported before (I see many articles claiming "firsts," when in fact it was simply the first for the authors), but based on the thoroughness of the literature cited, and some back-checking of my own, I do indeed believe this to be the first record in this extremely important fish family.

One question for Editorial staff: is reference to an article in prep acceptable? Not all journals allow it.

Experimental design

While studies of this type do not lend themselves to manipulation, I found the sample size and breadth in this study to be exceptional. The Methods were clear, the statistics rigorous, and the underlying premise testable (through significant differences).

Validity of the findings

I found the summary conclusions of the paper to be spot-on - sexually dimorphic olfaction was proved and its relation to taxon rarity in nature would indeed be the most parsimonious explanation.

The comments that I would consider substantive are as follows, with the proviso that these are possible improvements (i.e., the opinions of a single reviewer) rather than criticisms.

The authors spend a great deal of real estate in this paper discussing bioluminescence, which is certainly relevant, but at times veers off point relative to the main thesis of this paper. One suggestion would be to trim this treatment, and instead delve more into the four hypothetical uses of olfaction (several of which are indeed treated, but in less volume than that of bioluminescence): 1) finding food (treated thoroughly in Discussion, but not as much in the Intro, where the paper's thesis is presented); 2) detecting predators and/or danger; 3) discriminating watermasses, and thus overall distribution (not treated); and, 4) finding mates. This is treated in the Discussion (beginning line 351), but more upfront treatment might make for a more fluid read.

The last suggestion would be mention of what we know, or do not know, about the species-specificity of pheromones in fishes, or any animal, for that matter. This is the true underlying hypothesis - Loweina can smell the difference in species and thus "home in" on conspecifics. Without any proof along these lines, mate recognition is truly just conjecture. Even if we know ZERO on this topic, it would make for a good "We need to know more about this fundamental feature of ecology" statement.

Additional comments

Minor questions/comments

Line 191: earlier in text the term nostrils was used. People use them interchangeably, but my understanding is that nostrils communicate with the esophagus/trachea, whereas nares do not (and thus, fishes have nares), but I may have this wrong.
Line 220: "notable" might be a better word than "unusual" (less subjective)
Line 238: While "parasitism" is the term conventionally used to describe male fusion to females in ceratioids, technically "parasitism" involves two species, by definition. Thus, ceratioid attachment might best be characterized as "chimerism," the presence of two or more cell lines with different genetic origin within the same organism (Rejduch, 2001; Rejduch et al., 2016).
Line 246: what about nemichthids? Their sexual dimorphism is remarkable - does this map to olfaction?

---

## Round 0.2 · accepted · Accept

I thank the authors for following through with all the comments and suggestions by the reviewers. The current version of the manuscript is ready for publication. If possible, I agree with reviewer 1 that Figure 2B could be further enhanced by the addition of lines to indicate measured length and width.

·

Basic reporting

The authors have satisfactory addressed all my concerns. One very minor suggestion from my side: The authors could add lines to the Figure 2B, on the olfactory epithelium, to clearly show how was measured the length and width. Thanks to the authors for their work and I hope to see more studies on the olfaction of these species !

Experimental design

no comment

Validity of the findings

no comment

·

Basic reporting

What was a nice paper to begin with is even nicer with the authors' revisions.

Experimental design

Same comment as before - I believe this has been done as well as it can be done.

Validity of the findings

Same as before - all aspects of Results presentation and Discussion are well posed and valid.

Additional comments

Very nice piece of work.